# Gaming for Training Voluntary Control of Pupil Size

Leonardo Cardinali [1,†] , Silvestro Roatta [1,*] , Raffaele Pertusio [1], Marcella Testa [2] and Cristina Moglia [2,3]

1. "Rita Levi Montalcini" Department of Neuroscience, University of Turin, 10125 Turin, Italy
2. ALS Centre, "Rita Levi Montalcini" Department of Neuroscience, University of Turin, 10126 Turin, Italy
3. SC Neurologia 1U, AOU Città della Salute e della Scienza di Torino, 10126 Turin, Italy
* Correspondence: silvestro.roatta@unito.it
† Current address: Department of Electronics and Telecommunications, Politecnico di Torino, 10129 Turin, Italy.

**Abstract:** Users can "voluntarily" control the size of their pupil by switching focus from a far target A (large pupil size) to a near target B (small pupil size), according to the pupillary accommodative response (PAR). Pupil size is governed by smooth muscles and has been suggested as communication pathway for patients affected by paralysis of skeletal muscles, such as in amyotrophic lateral sclerosis (ALS). We here present a video game that relies on PAR: a 2d side-scroller game where the user, by varying pupil size, controls the height at which a spaceship is moving aiming at colliding with bubbles to burst them and score points. The height at which the spaceship flies inversely depends on pupil area. The game is implemented on a Raspberry Pi board equipped with a IR camera and may record the time course of pupil size during the game, for off-line analysis. This application is intended as a tool to train and familiarize with the control of pupil size for alternative augmentative communication.

**Keywords:** pupillary accommodative response; video game; human–computer interface; locked-in syndrome





## 1. Introduction

*Pupil in Software Control*

Studies show that augmentative and alternative communication (AAC) devices are fundamental to many people affected by motor diseases and improve the life of both patients and caregivers [1–3]. In particular, amyotrophic lateral sclerosis (ALS) is a major paralysing disease, with an incidence of 3–4/100,000 persons per year. It mainly affects the motor pathways of the nervous system (upper and lower motorneurons) which ultimately innervate all skeletal muscles of the body, responsible for the performing force and movement [4,5]. The progression of the disease is associated with progressive paralysis and impairment of voluntary movements as necessary for writing, speaking, and autonomous breathing, thus also making it difficult to communicate with the external world. Patients in an impaired condition because of ALS or other illnesses typically rely on eye-tracking systems to compensate their communication losses [6]. This technology has been around for decades now and is a robust method for software interaction allowing paralyzed people to communicate, work, and play games only using eyes movements. However, ALS eventually affects also extra-ocular muscles and eye movements become more and more limited. The patient reaches a locked-in state (LIS) when there is total immobility except for blinking and eye movements while complete locked-in state (CLIS) happens when there is no movement left at all [7]. In these cases, eye-tracking systems become unusable and new strategies have to be found. In the range of new possibilities there are brain–computer interfaces (BCIs) based on the electroencephalogram (EEG), on near-infrared spectroscopy (NIRS) and also on blood oxygenation level-dependent (BOLD) signal using fMRI [8,9]. In addition, alternative methods have been implemented to exploit efferent pathways of the autonomic nervous system, e.g., by monitoring skin conductivity, heart rate, and pupil size, which then need to be modified by the subject's will, according to different covert

strategies, e.g., by self increasing the state of arousal by mental calculation [10]. The option of considering autonomic pathways is particularly appealing because they appear to be relatively preserved in ALS. In particular, no ALS-related deficit in pupil function has been described [11,12] while no difference in basal pupil size and quantitative pupillometry was reported in ALS patients compared to matched controls [13]. We recently devised a simple and effective strategy to voluntarily reduce the pupil size [14], which exploits the pupil accommodative response (PAR) [15,16]. The PAR is the pupil constriction/dilatation that accompanies the increase (decrease) in curvature of the lens and the convergence (divergence) movement of the eyes that takes place whenever we shift the focus from a far to a near (from a near to a far) visual target, the combination of these three actions being known as triad [17]. Thus, a consistent pupil constriction can voluntarily be obtained by shifting the focus from a far to near target. Note that eye movement (convergence) is not strictly necessary, provided that the two targets are aligned along the gaze of the relevant eye. This can be accomplished by using a semi-transparent near target, allowing for simultaneous view of the far and near targets [14]. The PAR is easily performed without requiring procedural learning nor extensive training. In addition, we observed that it is well preserved in ALS patients, including those in LIS [18]. As for the implemented technique, it has proven adequate for developing AAC devices [18] although it has not yet been successfully tested in patients in CLIS. Since a rapid cognitive decline is considered to take place at the onset of the CLIS condition and contribute to the failure of subsequent attempts to re-establish a bidirectional communication with the patient [19], it is likely that adopting and training a new communication channel in due time may prevent the cognitive decline and help to maintain the channel active beyond the loss of eye movement. Adoption of gaming is a well known strategy to stimulate the patient engagement in the training/rehabilitation process [20] also with regard to eye movements [21]. To our knowledge only one report is present in the literature in which pupil size was exploited in a gaming application [22]. However, as discussed later more in depth, this application employed emotion-related pupil dilatation [23], which is difficult to self-activate. On this basis, the present application was specifically developed for the PAR, with the aim of creating a tool adequate to train this pupillary function and possibly preserve it across the LIS-CLIS transition.

The game has been created as an add-on for a low-cost pupil-size driven AAC device, which implements a binary communication for providing YES/NO answers to posed questions or for selecting one of several items on a screen through a scanning-selection approach [18].

## 2. Materials and Methods

### 2.1. Hardware

The first implementations of a PAR-based BCI made use of an eye tracker for monitoring pupil size and of a PC for data acquisition and processing [14,18]. With the aim of developing a stand-alone device that could be used by patients and caregivers with no need of specialized personnel, a new platform was recently developed based on the single board computer (SBC) connected to a a small IR camera [24].

The game runs on a similar platform: a low-cost and low-consumption SBC to which all other pieces of hardware are connected Figure 1. The system components are:

- *SBC Raspberry Pi 4 model B, 4 GB RAM (Raspberry Pi Foundation, Cambridge, UK).* The previous platform was based on a Raspberry Pi 3 [24], we chose the fourth generation for the improved computational power, the double microHDMI video output, useful to connect both the touch screen and the external monitor needed for the game, and the low cost (75 USD);
- *32 GB microSD.* Needed for the SBC to load the operative system;
- *2.0 MP 1080p USB IR camera (Shenzen DingDang Smart Technology Co., Ltd., Shenzen, China).* This camera is the same used for the previous platform [24];
- *IR led.* Used to enhance pupil contrast;

- *5 inch hdmi display xpt2046 touch controller (Elecrow)*. The resolution and the precision of this screen make it suitable for menu navigation using virtual buttons (Figure 1b);
- *2 speakers (Reyes 8002)*. Connected exploiting pulse width modulation (PWM) pins on the Raspberry Pi 4, the speakers dimension allow them to fit inside the 3D-printed case;
- *3D printed structure mounted on a cut eyeglasses frame*;
- *3D printed case*. Since the device is developed be portable, a compact 3D-printed case was used Figure 1;
- *MicroHDMI-to-HDMI cable*. Used to connect the device and an external monitor

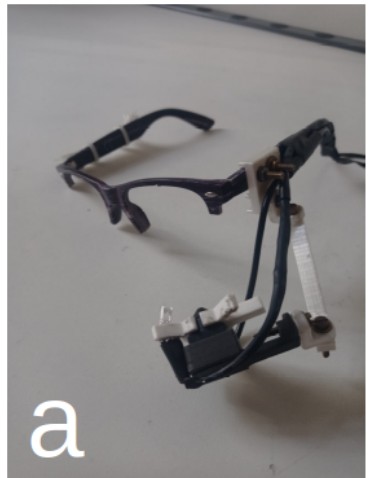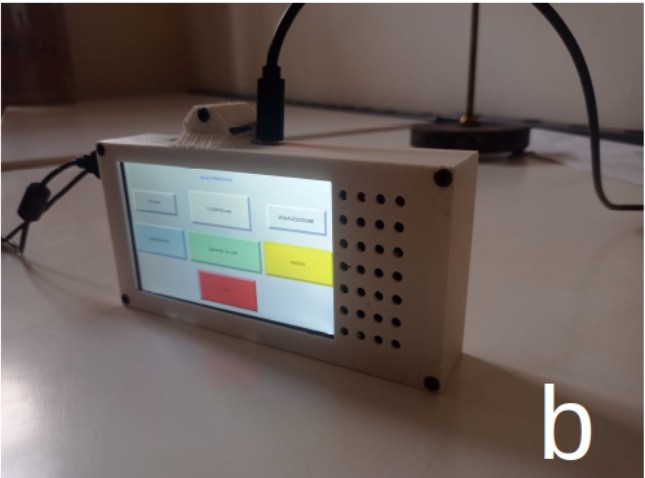

**Figure 1.** The device: (**a**) eyeglasses frame with IR camera and led, held by an adjustable 3D-printed support; (**b**) Raspberry Pi 4 equipped with touch screen and speakers, all embedded in a 3D-printed case.

The IR camera and IR led were mounted over a modified eyeglasses frame by means of a dedicated 3D-printed structure, presenting 5 degrees of freedom for easy adjustment of the position and orientation of the camera relative to the subject's left eye, to compensate for possible individual differences. The IR led is fundamental to enhance the contrast between the pupil and the iris and to reduce external light reflections, enabling pupil detection. The Raspberry, the touchscreen, and the speakers are embedded in a 3D-printed case, which makes available all slots for USB connections to the camera, the PC, etc., as well as a HDMI connection to larger video monitors, if necessary. The cost of the whole system is under EUR 200.

### 2.2. Software

The software is completely based on open source platforms. The operative system running on the Raspberry Pi is Raspbian GNU/Linux 10 (buster), a Linux distribution based on Debian, installed on a 32 GB micro SD card. The programming language used is Python 3.7.4, following an object-oriented distributed approach. The most important python modules used for the game creation are OpenCV (ver. 4.5.2), Turtle (module based on tkinter 8.6), and vlc (ver. 3.0.12118). OpenCV is a cross-platform open-source library for image processing that allowed us to process the frames collected by the IR camera extracting the pupil area value. Turtle is a graphic tool to draw shapes on a virtual canvas, the visual part of the game has been built entirely using this library. Vlc is an open source media player that we employed to play audio cues.

#### 2.2.1. Pupil Size Monitoring

To improve the performance of the algorithm, an automatic pupil detection is carried out in order to identify the region of interest (ROI) of the image where the pupil is located. Using the ROI instead of the whole image taken by the camera reduces the risk of false

pupil detection and considerably lowers the computational load of the subsequent video processing. The detection algorithm finds the pupil in the whole image (640 × 480 px) using the following steps:

1. Grayscaling;
2. Binarization;
3. Morphological dilatation;
4. Filling of closed contours;
5. Connected components analysis where pupil selection is based on:

   - Connected component mean value,
   - Connected component dimension,
   - Connected component eccentricity.

If the pupil is successfully found for 15 consecutive frames, the pupil centroid position is saved and used in the pupil size monitoring algorithm as the central point of the ROI (that will be a 320 × 240 px image), the pupil detection phase then stops and the main menu is shown to the user, that can now start the game. When the game is started the camera begins to collect frames. For each frame the ROI is cropped using the coordinates found during pupil detection and is then computed by an algorithm based on the ellipse fitting method described in the work from Romaguera et al. [25]. The steps of the pupil size monitoring algorithm are the following:

1. Grayscaling;
2. Thresholding;
3. Morphological closing;
4. Edges detection using Canny method;
5. Connected component analysis to detect the largest connected component, that we presume is the pupil contour;
6. Find the ellipse that best fits this contour;
7. Compute ellipse area.

Thus, for each frame, the pupil size is calculated as the number of pixels included in the ellipse area.

The choice of having different algorithms for initial pupil detection and for pupil size monitoring is due to fact that the initial pupil detection has lower chances of false positives but also lower sensibility while the other has higher sensibility and robustness but is also more prone to false positives. During detection the more sensitive method is used to have higher chances of correctly finding the pupil and selecting the correct ROI while during size monitoring where performances are increased due to ROI selection the more sensible method is used.

### 2.2.2. How the Game Works

The game is a 2D side-scroller set in space where the player controls the vertical movement of a spaceship to burst white bubbles in order to score points. The goal is to burst a certain amount of bubbles in the shortest possible time.

The horizontal position of the spaceship is fixed on the left side of the screen while bubbles spawn on the right side and move horizontally towards the left at fixed speed. The spawning times and heights of groups of bubbles are randomized. The player can move the spaceship vertically by changing the pupil size: constricting the pupil moves the spaceship upwards and vice versa. Voluntary control of pupil size is achieved through the pupillary accommodative response whereby the pupil constricts when shifting the focus on from a far to a near visual target. Thus, the player normally looks at the screen (far target) where the game is running, waiting for the bubbles to arrive. In this condition the pupil area is large and the ship stays on the bottom. When bubbles are close to pass over the spaceship the user may raise the ship and hit the bubbles by constricting the pupil (as shown in Figure 2) and may achieve this by transiently shifting the focus on the near target and subsequently returning to the screen.

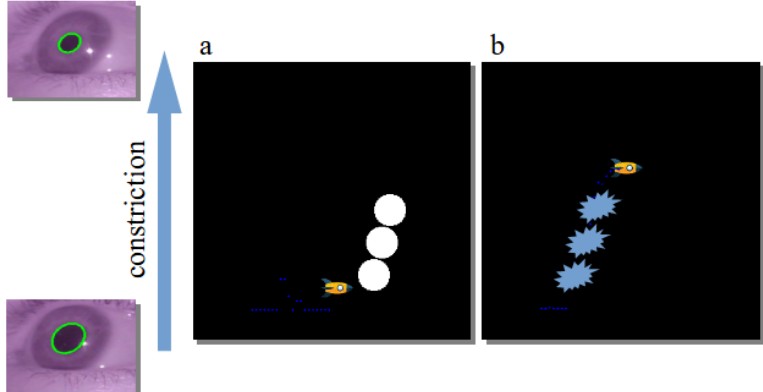

**Figure 2.** Screenshots of the game. The vertical position of the spaceship is related to the extent of pupil constriction: when looking at the far target (the display) the pupil is dilated, which leaves the spaceship on the ground (**a**). When transiently shifting the focus on the near target the pupil constricts, thus raising up the spaceship towards the incoming bubbles (**b**).

The near target is a transparent plastic surface, with small white dots depicted on it, located at about 30 cm from the eye of the subject, along the gaze line from the left eye to the screen. In this way, the subject can easily change the focus from the screen to the near target, with minimal or absent convergence movement. When the bubble is hit by the spaceship it explodes, the score is increased by 1 point and an acoustic cue is played. The audio signal produced by the bubble explosion (in case of successful hit) partly compensates the transient visual loss of the game scene while focusing on the near target (generally for about 1 s).

When the score reaches the value of 20, the game stops and and the subject performance is inversely related to the time employed to complete the game. Time scores are saved in a file and the top ten shortest times can be read from the game menu.

### 2.2.3. Flowchart

Here, we present the flowchart algorithm of the game:

The algorithm starting point (START in Figure 3) corresponds to starting the game from the device menu (Figure 1b). When selected, two threads start running: one for monitoring the pupil area and computing the spaceship vertical position $y$ (left branch) and the other for drawing each frame of the game (right branch). Both threads need an initialization phase before entering the respective loops, the timer that will be used for the player score is started when both thread end their initialization phase and the player is in control of the spaceship. The drawing thread, for each cycle, draws the spaceship in the $y$ position, read from the vertical position computing thread, then checks the timer to spawn bubbles at semi-random intervals in the right side of the screen, moves bubbles one step towards the left (the step length, in pixels, determines the bubbles speed) and deletes them if they arrive to the left side of the screen or if they collide with the spaceship. In case of a collision, the player score is increased by 1 point for each bubble that collided with the spaceship. After managing the object on screen and, in case of a collision, updating the score, the game-over conditions are checked. If these conditions are not met, the frame is drawn and displayed on screen, then the next objects positions and scores are computed and the loop repeats. If, instead, these conditions are met, the game will end both threads, draw a final frame where the total time is displayed and return to the device main menu. The game-over conditions are met when the score (burst bubbles) reaches 20 or when the touchscreen detects a tap in any point. In the first case, the time of the last drawn frame is saved in a high scores table, in the second case the time is not saved, since this second condition ends the game prematurely. The thread computing vertical position $y$ will be described in the following section.

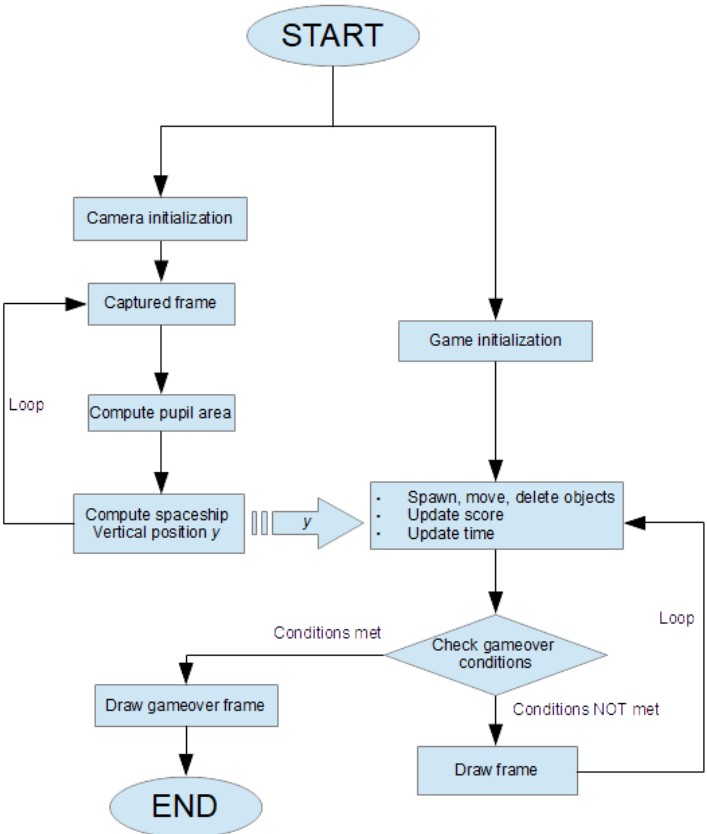

**Figure 3.** Game algorithm flowchart.

2.2.4. Converting Pupil Size to Spaceship Vertical Position

The first step is pupil detection capturing a frame through the IR camera and processing it through the segmentation algorithm, then a robust and adaptive algorithm has been devised to vertically locate the spaceship, based on the current size of the pupil. The spaceship vertical position $y$ is calculated using the following equation:

$$y = Y_M - \frac{a - A_M}{A_M - A_m}(Y_M - Y_m) \tag{1}$$

where $a$ is the current pupil area obtained from the pupil detection algorithm, $Y_M$ and $Y_m$ are, respectively, the upper and lower limits of the spaceship possible vertical position and $A_M$ and $A_m$ are, respectively, the largest and smallest area values detected. All values are in pixels. Although $Y_M$ and $Y_m$ are static values, $A_M$ and $A_m$ are dynamic to improve the robustness and reliability of the algorithm. $A_M$ is initialized to a slightly smaller value than the pupil area value found when the game is started and $A_m$ is initialized to a slightly bigger value. Whenever an area value larger than $A_M$ is acquired, that value becomes the new $A_M$ and when an area value smaller than $A_m$ is acquired, it becomes the new $A_m$ value. To avoid erroneous pupil size calculations from pushing these limits to numbers that would make the algorithm faulty, $A_m$ constantly increases while $A_M$ constantly decreases by two quantities found empirically so that the effect of the outliers neutralizes in a reasonable amount of time.

Once the $y$ is computed, the value is saved and it can be read by the drawing thread when it needs to draw the spaceship, and the loop repeats by capturing another frame from camera.

### 2.3. Tests on Healthy and Patients

Our subject pool consists of 11 healthy subjects of age between 21 and 23 years old (5 males and 6 females) and 3 patients affected by ALS in late stage (with a locked-in syndrome).

The research was conducted in accordance with the ethical standards of the 1964 Helsinki declaration and its later amendments or comparable ethical standards and under the approval of the Ethics Committee of University of Turin (prot. 256,076). An informed consent was obtained from the subjects and/or their caregivers.

For test sessions the subjects were first of all informed about how the system and the game work and how to control the pupil size using PAR. The informative document was read entirely and the informed consent document was signed by both the experimenter and the subject. The subjects then sat comfortably at 150 cm from the far target (the screen where the game is displayed) and at 20 cm from the near: the two targets being roughly aligned along the gaze line of the left eye so as to minimize eye movements when shifting the focus between them (Figure 4). These distances have been found to be optimal during empirical tests in the developing phase. Finally, the eyeglasses frame that holds the camera was worn and the device was started. Before the game, the subjects were asked to perform a focus-shifting task consisting in shifting the focus from the far to the near target, according to an audio cue, maintaining the focus on the near target for the duration of the cue and then returning on the far target until the next cue. Six cues were presented, lasting 2, 4, 6, 6, 4, and 2 s, separated by 10-s intervals.

Three rounds of the game were then played. After the game, the focus shifting task was repeated. Pupil size was monitored and saved in a file for the two focus shifting task and for each round of the game for further analysis. After the experiment, the subjects were asked to answer four questions:

1. How much did you feel in control of the spaceship on a scale from 1 to 5?
2. How much fun did you have on a scale from 1 to 5?
3. How much effort did the game require on a scale from 1 to 5?
4. Did you perceive yourself improving during the rounds?

For ALS patients the distance from the far target was longer for the constraints due to their condition, but always less than 3 m. This increased distance does not meaningfully affect the PAR, and other studies from our group use a similar distance between the subject and the far target [14].

The clinical conditions of the three patients are now briefly described:

- Patient A: 56 years old, male, was having problems using current eye-tracking device since many months at the time of the test;
- Patient B: 34 years old, female, has been using an eye-tracker communicator for three years and still uses it with ease at the time of the test;
- Patient C: 52 years old, male, was having problems using current eye-tracking device since many months at the time of the test;

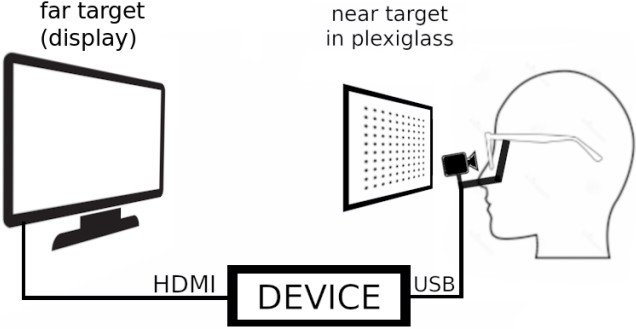

**Figure 4.** Experimental setup.

## 3. Results

### 3.1. Healthy Subjects

In two female subjects the pupil-detection routine did not provide a reliable and continuous tracking of the pupil, due to dark eye make-up. The analysis was, therefore, carried out on the remaining 9 subjects who easily understood the instructions and completed the focus-shift task and the three rounds of the game without problems.

The recording of pupil size from a representative subject is shown in Figure 5. On average, the magnitude of pupil size constriction during the game was 49.6 ± 7.9% of the maximum value.

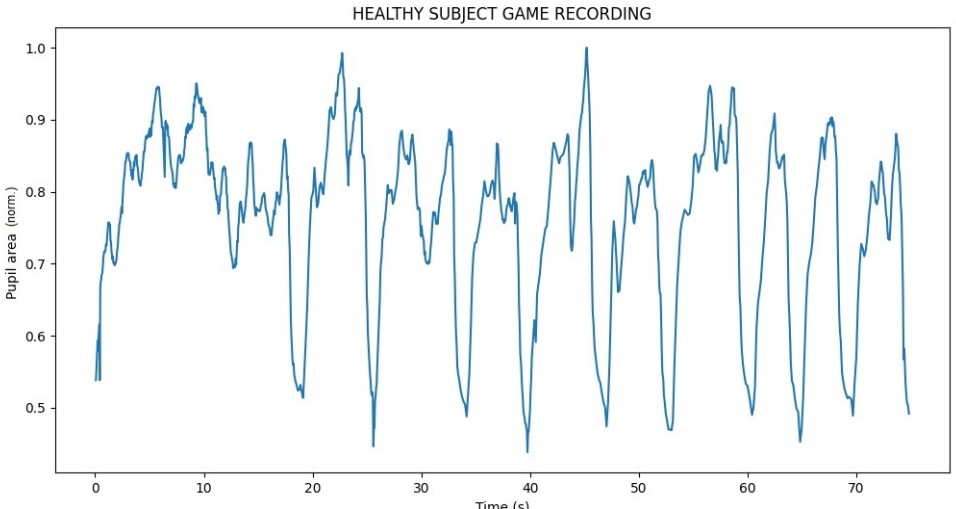

**Figure 5.** Time course of pupil size recorded during a game in a representative subject. Pupil area was normalized to the maximum recorded value. Note the several distinct constrictory events.

The results of the survey are presented in Figure 6 and detailed below:

- *Control* : none of the subjects felt perfectly in control of the ship (score: 5/5) but also none felt completely out of control (score: 1/5).
- *Effort*: this is the parameter with the highest variability. Some of the subjects easily managed to raise the ship at their will while for others several attempts were necessary to correctly and timely activate the PAR.
- *Fun*: no subject gave the lowest score to this parameter, but while some subjects found the game interesting others found it a bit boring due to the monotony and low repeatability of the task, even if the bubbles spawn at random places and time intervals.
- *Perceived improvement*: improvement was not numerically quantified, but all subjects reported that the control of the spaceship progressively improved during the repetitions of the game.

In Figure 7 we can see the box plot of the scores of all healthy subjects in the three matches of the game. However, the actual performance, assessed in terms of time required to complete the game, does not exhibit a net trend of improvement.

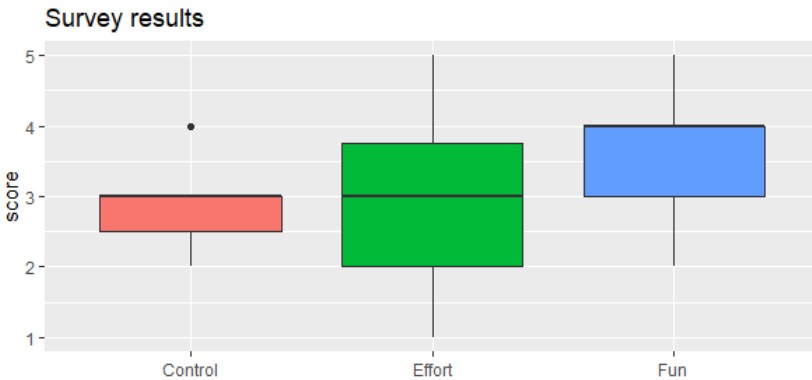

**Figure 6.** Box plot representing survey results in terms of (1) perceived control of the spaceship (i.e., of pupil size); (2) perceived effort in controlling the spaceship; and (3) fun experienced during the game. Subjects were required to score each variable with a number ranging from 1 (minimum) to 5 (maximum). N = 9. Box plots represents the data range in quartiles, the interquartile range (IQR) which is Q1 to Q3 is represented in the box, the thicker horizontal line is the median value, vertical lines indicate the maximum and minimum values, outliers are represented as dots.

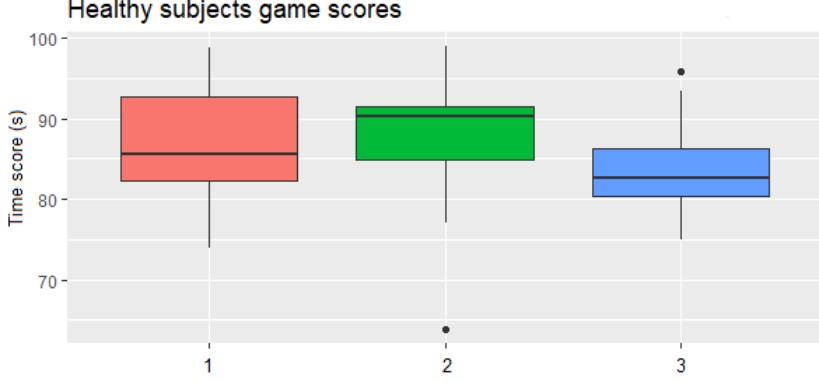

**Figure 7.** Performance achieved by healthy subjects in the three repetitions of the game, quantified in terms of time required to complete the game (explosion of 20 bubbles). Indications as in Figure 6.

### 3.2. Patients

Patient A completed the game rounds but it is not clear if he was in control of the ship, since the pupil did not exhibit clear-cut, detectable constrictions during the focus shifting task nor during the game. Patient C was too tired and could maintain attention only for short periods of time so that it could not be engaged in the preliminary task nor in the game. Patient B effectively completed the focus shifting task and the game rounds achieving results comparable to healthy subjects, as described below. The magnitude of pupil size constriction during the game was on average 45.6 ± 10.8 % of maximum value (Figure 8).

Patient B gave the following scores for the survey parameters:

- *Control*: 3 out of 5.
- *Effort*: 2 out of 5.
- *Fun*: 5 out of 5.
- *Perceived improvement*: Yes.

Patient B scores were the following: 114.69 s for the first match, 88.90 s for the second match, 66.29 s for the third match. After understanding the basics of the game and how the spaceship responded to focus shifting, the patient quickly improved the score performing much better than most of healthy subjects in the last match.

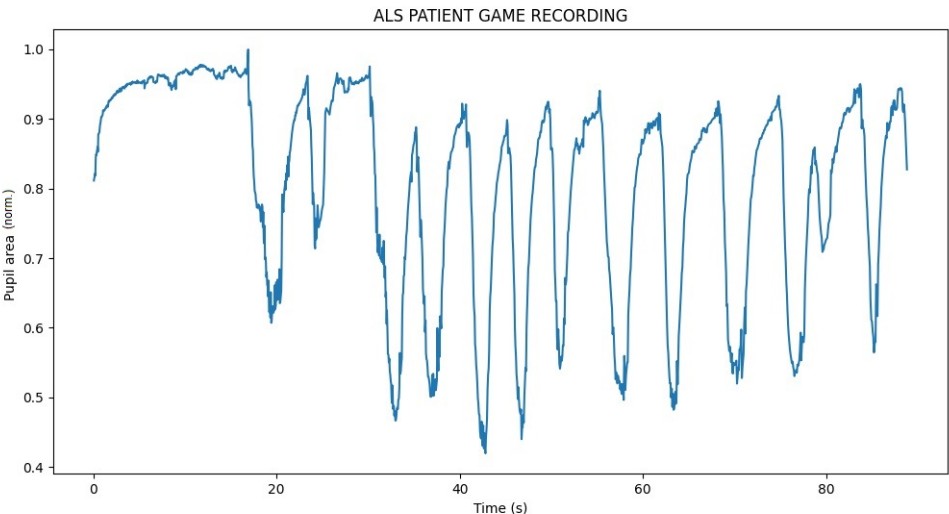

**Figure 8.** Time course of pupil size recorded during a game in Patient B. Pupil area was normalized to the maximum recorded value. Note the several distinct constrictory events of magnitude comparable to those of the healthy subject, reported in Figure 5.

## 4. Discussion

In this work a simple pupil-driven game, implemented in a low-cost AAC device prototype, was presented and tested in a group of healthy subjects and in three late-stage ALS patients. The results indicate that subjects can immediately and successfully control pupil size and effectively play with the device with some fun, virtually in the absence of any movement. An excellent performance was also achieved by a late-stage ALS patient. In spite of the not impressive wide diffusion and availability of eye-tracking devices with multiple applications, particularly in the field of AAC, only few studies explored the possibility to employ the changes in pupil size as a communication pathway [10,14,18,26–28] and only one study described the implementation of pupil size to control a game [22]. In this latter conference paper, eye movements are used to move the game character (the girl Eni) around while pupil dilatation is used to enforce the opening of magic flowers (as necessary to set butterflies free, which is the objective of the game). To achieve pupil dilatation while looking at the flower, the player is required to self-activate arousal or "positive emotions", which increases the sympathetic drive to the pupil dilator muscle. However, as acknowledged by the authors, "trials with novice users indicate that pupil control is hard, if not impossible" and that extensive training in controlling and provoking one's emotional status, while receiving feedback of current pupil size, was necessary to acquire the capability to evoke pupil dilatations [10]. Possibly due to this reason, this approach was not followed-up, nor in gaming nor in other AAC applications. Conversely, the current implementation is based on a different way to control pupil size, the PAR, which produces sharp and clear-cut pupil constrictions that can easily be realized by most subjects with minimal/moderate effort and without any prior training. In fact all subjects were tested on their first session with the device. The little necessity of a learning phase is also demonstrated by the average lack of/or moderate improvement in performance during the three subsequent matches. This characteristic is particularly important when aiming at using AAC devices with critical patients in the LIS or CLIS condition.

We were happy to observe a prompt and effective engagement of patient B who also exhibited excellent performance, comparable to the young and healthy population, and her enjoyment in playing the game. This is a major outcome of this study because it means that this game (and future improved versions or different implementations) may be a viable tool to engage the LIS patients in using the PAR while it is still functional, which would arguably help in maintaining this functionality when the progression of ALS will

move them towards the CLIS condition. In fact, we cannot exclude that worsened eye functionality and cognitive conditions might have contributed to the failure in using PAR in patients A and C.

During the game development, pupil area has been used in different ways to compute the spaceship vertical position. Since the device already had functions to detect pupil constriction events, one of the approaches was to have the spaceship constantly falling towards the bottom boundary while the constriction event was not detected, and constantly rising at fixed speed while it was detected. This was a viable solution, but we found the game less interesting and less informative about PAR with respect to the spaceship position being a direct representation of the current pupil size. The game is intended for learning and rehabilitation purposes, to enable pupil size control using PAR as a tool for communication and software interaction. Having continuous and detailed feedback is crucial for any learning process [22]. For this reason, we chose to directly link the spaceship position to pupil size. This presented many challenges, first of all the definition of pupil size boundaries: it is very important for the player to be able to reach the top, as well as the bottom of the screen, and fixed $A_M$ and $A_m$ values (Equation (1)) failed to adapt to the ambient noise that influences pupil dimension, such as changes in room brightness and thought-driven dilatation and constriction. Dynamic $A_M$ and $A_m$ values allow the equation to be more flexible and to adapt to different situations, do not need the player to have a calibration phase each time the game is started and, during the development phase, gave better feedback. The main drawback is an enhanced sensibility to false pupil values: excessively big or small values due to false pupil detection or gaze movements can alter the correct interpretation of the pupil size, leading to the necessity of restarting the game. For this reason the pupil detection algorithm has to be robust and reliable. Two subjects were tested but excluded from the analysis because of the low correct pupil detection rate. This problem was due to the dark make-up the subjects wore, which tricked the detection algorithm. We believe that with more sophisticated pupil detection systems, as those commercially available, this problem would disappear.

The result of the survey also deserves some consideration. We can observe that the perceived control of spaceship was not very high and, at the same time, required a non-negligible effort (Figure 6). This may be attributed to the fact that the subjects, at their first approach with the device, were still little confident with the voluntary pupil control. In addition, the emotional engagement associated with the novelty of the game could have introduced spurious alterations of pupil size [23]. The three repetitions of the game were held in a short time, within the same session, and with no significant improvement in performance (Figure 7). This however may be due to a ceiling effect, i.e., the spaceship control was already good enough to get close to the maximum possible performance. Notably, the time course of pupil size during the game is recorded, as shown in Figures 5 and 8, and can be later accessed for analysis. This is intended as a tool to monitor the characteristics of PAR (e.g., the magnitude and the speed of pupil constriction) and their trend during the progression of the disease.

## 5. Conclusions

In conclusion, in this study a new game is presented that can be played without a single body movement, being exclusively controlled by changes in pupil size. The game has been successfully tested by a group of healthy subjects and an ALS patient in LIS. The results provide a proof viability of this approach as a possible tool to engage and motivate patient in LIS to train and maintain functional their pupillary function in the eventual transition to CLIS.

**Author Contributions:** Conceptualization, S.R.; methodology, S.R., R.P. and L.C.; software, L.C.; validation, L.C., M.T. and C.M.; formal analysis, L.C.; investigation, L.C., S.R. and M.T.; data curation, L.C.; writing—original draft preparation, L.C. and S.R.; writing—review and editing, S.R.; visualization, L.C.; supervision, S.R.; resources, R.P. project administration, S.R.; funding acquisition, S.R. All authors have read and agreed to the published version of the manuscript.

**Funding:** This study was supported by the University of Torino (Proof of Concept, TOIMPROVE/2020).

**Institutional Review Board Statement:** The study was conducted in accordance with the Declaration of Helsinki, and approved by the Ethics Committee of the University of Torino (protocol code 256076 and 21 July 2017 of approval).

**Informed Consent Statement:** Informed consent was obtained from all subjects involved in the study. Written informed consent has been obtained from the patient(s) to publish this paper.

**Data Availability Statement:** The data presented in this study are available on request from the corresponding author.

**Conflicts of Interest:** The authors declare no conflict of interest.

## Abbreviations

The following abbreviations are used in this manuscript:

| | |
|---|---|
| AAC | augmentative and alternative communication |
| ALS | amyotrophic lateral sclerosis |
| BOLD | blood oxygenation level-dependent |
| BCIs | brain-computer interfaces |
| CLIS | complete locked-in state |
| EEG | electroencephalogram |
| fMRI | functional magnetic resonance imaging |
| HR | heart rate |
| IQR | interquartile range |
| IR | infra-red |
| LIS | locked-in state |
| NIRS | near-infrared spectroscopy |
| PAR | pupillary accommodative response |
| PWM | pulse width modulation |
| ROI | region of interest |
| SC | skin conductivity |

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
