# Peer review of "Gaming for Training Voluntary Control of Pupil Size"

_electronics, doi:10.3390/electronics11223713_

Round 1
Reviewer 1 Report
I have the following remarks:
The paper is interesting and deals with an important problem for people affected by motor diseases.
Explain more clearly the Game algorithm flowchart from Fig. 3.
Develop in Section1 a comparison with similar methods from the literature.
What real applications can use the proposed method?
Author Response
I have the following remarks:
The paper is interesting and deals with an important problem for people affected by motor diseases.
Develop in Section1 a comparison with similar methods from the literature.
ANSWER: To our knowledge there is only one report in the literature describing gaming application based on the control of pupil size. It was already considered in the discussion but a reference has now also been added in the introduction.
What real applications can use the proposed method?
ANSWER: This is actually already an application of the PAR-based communication methodology. This implemented game is just a proof o concept showing that PAR can be used also for gaming. At the same time it suggested as a way to train the PAR and possibly “keep it alive” in the transition from the LIS to the CLIS condition, as may occur to ALS patients. The issue has been further emphasized in the conclusion.
Reviewer 2 Report
How can you measure the Pupil size of the eye.
What is the difference between the ALS and normal persons?
Any difference in the pupil size between the ALS and normal persons?
The study was conducted in online or offline mode?
Need more explanation for PAR.
Why did you give 5 degrees of freedom adjustments?
Explanation for the hardware implemented in the study was not sufficient.
What is the need of implementing automatic pupil detection algorithm? Can you try with some other algorithm?
Figure.2 a and b were not clear.
Subject selected for this study was not sufficient (5 males and 6 females) and 3 patients affected by ALS.
Can you measure any difference between normal and ALS affected person.
Figure.6, 7, 8 interpretation was not sufficient.
Need to improve the Result percentage by altering some of the parameters.
References was not sufficient in this paper.
Conclusion of the study was not present in the paper.
Author Response
How can you measure the Pupil size of the eye.
ANSWER: After the pupil has been identified by the algorithm and tracked frame-by-frame, for each frame the pupil size is quantified as the number of constituent pixels. This aspect is now better emphasized in section 2.2.1 (L. 181)
What is the difference between the ALS and normal persons?
ANSWER: A new paragraph was added in the Introduction to briefly introduce ALS disease (L 18)
Any difference in the pupil size between the ALS and normal persons?
ANSWER: The following sentence was added in the Introduction: “The option of considering autonomic pathways is particularly appealing because they appear to be relatively preserved in ALS. In particular, no ALS related- deficit in pupil function has been described [Rojas 2020; Cozza 2021] while no difference in basal pupil size and quantitative pupillometry was reported in ALS patients compared to matched controls [Baltadzhieva 2005]” (L. 39)
The study was conducted in online or offline mode?
ANSWER: In order to “play with the pupil”, every step (image acquisition, image processing, pupil identification and measurement, shift of the spaceship upward/downward according to pupil size) had to be conducted in real time. Measurements about the extent of pupil constrictions achieved during the game were performed off-line, for statistical purposes.
Need more explanation for PAR.
ANSWER: The explanation of the PAR as a component of the triad, along with relevant references has been added to the introduction (L 46)
Why did you give 5 degrees of freedom adjustments?
ANSWER: The relevant sentence has been integrated as follows: “The IR camera and IR led were mounted over a modified eyeglasses frame by means of a dedicated 3d-printed structure, presenting 5 degrees of freedom for easy adjustment of the position and orientation of the camera relative to the subject’s left eye, to compensate for possible individual differences.” (L 100)
Explanation for the hardware implemented in the study was not sufficient.
ANSWER: The referee is correct in pointing this out. We integrated additional details about the hardware whose full description is contained in another manuscript, currently under review [Chiarion et al, submitted]. (L76)
What is the need of implementing automatic pupil detection algorithm? Can you try with some other algorithm?
ANSWER: The automatic pupil detection is a very convenient feature of the package. In a previous version the initial identification of the pupil had to be performed manually by the operator. This can certainly be obtained also by other algorithms, but since we were quite happy with the achieved performance we did not test alternative solutions.
Figure.2 a and b were not clear.
ANSWER: Thank you for pointing this out; the figure has been rearranged and hopefully made more clear
Subject selected for this study was not sufficient (5 males and 6 females) and 3 patients affected by ALS.
ANSWER: We agree that in order to describe the pupil response in healthy subjects or patients a larger number of subjects and patients should have been collected. However please note that the PAR per se has already been described in healthy subjects and patients (new refs have been added and one study still in progress, specifically investigating ALS patients with a longitudinal and cross-sectional recruitment has been published in abstract form [Roatta et al 2019]) and that the focus of this study is to present and describe a new gaming application.
Can you measure any difference between normal and ALS affected person.
ANSWER: Although comparison of ALS patients and healthy control is beyond the scope of the present study, we acknowledge that this information was missing, It is now mentioned in the introduction, along with the relevant ref, that ALS patients, including those in the LIS condition have a normal PAR. (L39)
Figure.6, 7, 8 interpretation was not sufficient.
ANSWER: Thank you for pointing this out. We realized that we did not adequately consider the results of the survey. The legends of the figures were expanded, and new comments were added in the Discussion. (L 375)
Need to improve the Result percentage by altering some of the parameters.
ANSWER: We are not sure to understand this remark. The only result expressed in percentage is the peak magnitude of pupillary constriction. The observed magnitude (49%) is not dissimilar to the one previously reported (21.5 % reduction in diameter, meaning 39.6 % reduction in pupil area) [Ponzio et al 2019], considering the different experimental conditions and protocol. In particular, the magnitude of constriction is affected by the duration of focussing on the near target.
References was not sufficient in this paper.
ANSWER: Several references were added to support previous and newly integrated text
Conclusion of the study was not present in the paper.
ANSWER: We apologize for omitting this part. Conclusions have now been added to the discussion (L388)
Round 2
Reviewer 2 Report
Research idea was good